# A Survey on the Rationale Usage of Antimicrobial Agents in Small Animal Clinics and Farms in Trinidad and Jamaica

**DOI:** 10.3390/antibiotics11070885

**Published:** 2022-07-01

**Authors:** Muhammad Sani Ismaila, Alexandra Thomas-Rhoden, Angel Neptune, Kezia Sookram, Samantha Gopaul, Travis Padarath, Anil Persad, Karla Georges, Venkatesan Sundaram

**Affiliations:** 1Department of Basic Veterinary Sciences, School of Veterinary Medicine, The University of the West Indies, St. Augustine, Trinidad and Tobago; alexandra.thomasrhoden@my.uwi.edu (A.T.-R.); angel.neptune@my.uwi.edu (A.N.); kezia.sookram@my.uwi.edu (K.S.); samantha.gopaul@my.uwi.edu (S.G.); travis.padarath@my.uwi.edu (T.P.); karla.georges@sta.uwi.edu (K.G.); venkatesan.sundaram@sta.uwi.edu (V.S.); 2Department of Clinical Veterinary Sciences, School of Veterinary Medicine, The University of the West Indies, St. Augustine, Trinidad and Tobago; anil.persad@sta.uwi.edu

**Keywords:** antimicrobials, CARICOM countries, mixed practice, small animal practice

## Abstract

The extensive and indiscriminate use of antibiotics is known to contribute to antimicrobial resistance. Unfortunately, there are no public records of antimicrobial use (frequency or dosage) administered to animals in two major CARICOM (Caribbean Community) countries: Trinidad and Tobago, and Jamaica. Surveillance would promote amendments and discussion on a Caribbean antimicrobial-use protocol. In this study, an online survey was conducted using cross-sectional qualitative interviews via email, targeting veterinary clinicians working in clinics and farms in Trinidad and Jamaica, to identify how antimicrobials are used in the two countries. Out of the thirty-two (32) clinicians interviewed in Trinidad, 22 (68.75%) were small animal practitioners, and 10 (45.45%) were mixed practitioners. While in Jamaica, a total of Twenty six (26) clinicians responded, of which 17 of them (65.38%) were small animal practitioners and nine (34.62%) were mixed practitioners. A total of 95.2% of clinics and farms in Jamaica and 87.1% in Trinidad did not use standard antimicrobial protocols, which could be due to the limited availability of resources. The broad-spectrum antibiotic, amoxicillin, and amoxicillin/clavulanic acid were the most commonly used drugs in small animal practices in both countries (71.9% and 53.8% in dogs), (78.1% and 65.9% in cats); amoxicillin is also used frequently in mixed animal practice in Jamaica (44.4% in goats, 33.3% in cattle and 22.2% in sheep and pigs), while procaine penicillin and streptomycin was the most frequently used in mixed practice in Trinidad (60% in cattle and goats, 50% in sheep), which could explain the potentially increased risk of antimicrobial resistance.

## 1. Introduction

The use of antimicrobials in the treatment of animals is an important segment of veterinary practice, which must be closely monitored and critically reviewed because of its association with resistance, drug interactions, and adverse drug reactions. The rational use of antimicrobials aims to prevent the epidemic spread of contagious animal diseases, ensure a high efficiency of animal production, protect animal welfare, prevent the transmission of zoonotic diseases from animals to the human population, ensure the safety of food of animal origin, and prevent foodborne diseases [1,2,3,4,5]. Antimicrobial resistance is a global public health challenge that has been accelerated by the overuse of antibiotics worldwide [6,7,8]. Antimicrobials are used in both human and veterinary medicine, either for prophylaxis or treatment of microbial infections. The use of these drugs has drastically reduced the mortality rate due to microbial infections. In animal production, these drugs are mainly used against microbial infections; prophylactically for protection or fattening of livestock [2,9]. The World Health Organization (WHO) has ranked antimicrobial resistance as one of the top ten global public health threats [10]. In recent years, it has been observed that companion animals may pose a risk for the spread and dissemination of antibiotic-resistant genes to humans, due to their proximity with humans and as the use of antibiotics in these animals is more similar to that in humans [10,11]. Indiscriminate use of antimicrobials in both food and companion animals can lead to a severe crisis and cause some microbes to develop resistant genes to various groups of antimicrobial drugs [12,13,14,15,16]. Some researchers reported that antimicrobials were the most abused, overused, and misused substances in veterinary medicine [17,18]. Despite the presence and implementation of global antimicrobial resistance control strategies in most developed countries, gaps still exist in terms of the ability to curtail and control the rates of usage of antimicrobial agents in food animal, proper surveillance for resistant organisms, and total implementation of the global plans to control AMR [19,20]. In Europe, although data on the sale of veterinary drugs are available, data on the number of antimicrobials used in different animal species are lacking [21,22,23,24,25]. In other countries such as India, reports on the sale of antimicrobials are available, but information on the use of specific doses of these drugs in different clinics is lacking [26,27]. Another study in Turkey, which explored the use of antimicrobial agents by cattle farmers, revealed that most of the farmers had inadequate information about antibiotics and that they used them inappropriately, and most of them considered antibiotics as having antipyretic and analgesic effects, while 64% of them took advice from other farmers about antibiotic use and 48% did not need to consult a veterinarian before antibiotic use [28]. A study of antimicrobial use in small animals in South Africa found that sensitivity testing was not performed on small animals before treatment with antimicrobials. Many clients treat their pets before they come to a veterinary clinic [29]. Bonelli et al., 2014, reported that South America is one of the regions having the highest rate in the world in terms of antimicrobial resistance in Entrobactericeae [30]. In the Caribbean, to the best of our knowledge, there is no documentation of published data on the pattern of antimicrobial use, appropriate pre-treatment susceptibility testing, and the general distribution of antimicrobial use in both livestock and companion animals. This study was aimed at identifying the extent of the antimicrobial treatment in the Caribbean by assessing the use of antimicrobials in two major Caribbean islands: Trinidad and Jamaica. This includes the types of drugs used, their frequency, and indications for use by veterinarians on the islands in question.

## 2. Results

### 2.1. Trinidad Questionnaire Analysis

Of the sample size of sixty-two (62) veterinarians calculated for Trinidad, thirty-two (32) clinicians responded. The results showed that the practicing veterinarians ranged in age from 26 to 50 years. The number of female respondents also exceeded the number of male respondents, with female veterinarians comprising almost 80%. Twenty-nine (29) of the veterinarians worked in a clinic that was certified by the International Organization of Standardization (ISO) and had a written antibiotic policy. Thirty-two (32) veterinarians were found to prescribe antimicrobials frequently, while nine (9) veterinarians prescribed combinations of different antimicrobials, to achieve an additional effect. Owner compliance challenges also appeared to be an issue among professionals, as they were encountered frequently during the visits.

The most commonly used drug in small animal practice was amoxicillin (71.9% in dogs and 78.1% in cats), while in mixed practice, procaine penicillin and streptomycin were the most frequently used (60% in cattle and goats, 50% in sheep) (Table 1). The reasons for choosing an antimicrobial were mainly based on experience, while the vast majority also relied on literature such as therapy manuals for indications and administration.

The results also show that in cases where antibiotics were required, microbiological analysis and antimicrobial susceptibility testing were not performed, but the vast majority of cases used empirical antibiotics while waiting for the antibiogram. It should also be recognized that 97.1% of respondents kept records of antibiotic prescriptions for each client. The remainder did not, due to lack of time or because they felt it was not important to do so. In addition, a significant number of veterinarians (80%) had animals returned for failed antibiotic treatment, which they attributed to owners’ lack of adherence to veterinarian-recommended dosages and intervals. In some cases (20%), this was also due to antimicrobial resistance to the most frequently used antimicrobials.

### 2.2. Jamaica Questionnaire Analysis

Of the 34 clinics in Jamaica, twenty-six (26) clinicians responded to our questionnaire. Their graduation year ranged from 1984 to 2020; 90% of them have a DVM degree only, 65.4% are small animal veterinarians, and 34.6% are mixed practitioners. Most were employed in a practice they did not own, in eight parishes throughout Jamaica. None of the clinics in which they practiced were certified by the International Organization of Standardization (ISO), and only one clinic had a written antibiotic policy in its practice. According to the responses, owner-initiated treatments were a problem and frequently observed, with respondents answering either “frequently” or “sometimes” when asked how often they encountered them. It was noted that the majority of respondents had no problems with owner compliance with treatment regimens. Twenty-four (24) of the 26 respondents reported frequently prescribing antibiotics and two (2) respondents did not frequently prescribe them. Most respondents frequently prescribed combinations of different antimicrobials (57.7%), with the main reason being to achieve an additive effect. The most commonly used antimicrobial drugs across all species and practices were amoxicillin, in dogs (53.8%); cats (65.9%); (44.4% in goats, 33.3% in cattle, and 22.2% in sheep and pigs), (Table 1). The most commonly cited routes of administration were oral, topical, and intramuscular, with respondents choosing the antibiotic based on either experience (42.3%), treatment manuals (34.6%), or availability (19.2%). Most used microbiological analysis and antimicrobial susceptibility testing before antibiotic use and frequently used a broad-spectrum antimicrobial while waiting for laboratory results. Most respondents also reported that animals were returned to their clinic because of antibiotic treatment failure, with the primary reason being antimicrobial resistance or improper dosing intervals or deviations from the prescription by the owners. Some respondents (15) confirmed that no adverse reactions occurred in animals they treated with antimicrobials, while other (11) respondents who experienced adverse animal reactions cited gastrointestinal distress as the first primary sign observed. Most of the respondents (38.5%) also use postoperative antibiotics. 

### 2.3. Comparative Analysis of Results—Trinidad and Jamaica

The predominant veterinary practices in the two Caribbean countries are small animal and mixed animal practices (Figure 1). The data also show that there was no significant difference between the number of years in practice and the choice of antimicrobial agent in both Trinidad and Jamaica (p = 0.9696 and 0.9987, respectively). 

Our study indicated that, the percentage of veterinarians employing antimicrobial combination therapy is higher in Trinidad (74.3%) compared to Jamaica (57.7%), (Figure 2).

There were several reasons why veterinarians in the two countries chose a combination therapy. However, the most common reason was additive and synergistic effects (Figure 3).

Our study also compared the association between the two CARICOM countries in terms of choice of antibiotic use in both small animal and mixed practice. For small animal clinics, there was no association between country and the choice of antibiotic used for any of the antibiotic classes investigated (Table 2). However, for mixed animal practices (Table 3), clinics in Trinidad were more likely to use Penicillin-based antibiotic combinations compared to clinics in Jamaica (*p* < 0.01). 

In terms of different practices, organ systems, and their antibiotic associate, it was found that in Jamaica, amoxicillin-clavulanic acid, a popular broad-spectrum antibiotic, was commonly used to treat skin, musculoskeletal, and reproductive conditions; while the trimethoprim–sulfamethoxazole combination was used for both gastrointestinal tract and the central nervous system infections (Figure 4). Amoxicillin-clavulanic acid proved to be a favorite among veterinarians. More than 49% of the respondents reported its use for skin, respiratory, musculoskeletal, genitourinary, reproductive, and sepsis conditions. It should be noted that, according to the data collected, no antibiotics were used on the central nervous system in Trinidad (Figure 4).

It has been observed (Figure 5), that, most of the clinics in Jamaica (96.2%) and Trinidad (82.9%) had no written antimicrobial policy whereas the rate of antimicrobial prescription is high in both the two Caribbean countries (Figure 6).

## 3. Discussion

Despite tremendous advances in biomedical research, antimicrobial resistance remains the greatest threat to public health [31]. Although there are antimicrobial resistance surveillance and monitoring programs worldwide, including the Danish Integrated Antimicrobial Resistance Monitoring and Research Program (DANMAP), the Food and Drug Administration (FDA) National Antimicrobial Resistance Monitoring System, and the Global Antimicrobial Resistance and Use Surveillance System, there is no program specific to the Caribbean. The relationship between companion animals and humans, concerning AMR, presents an increased risk of cross-infection with zoonotic pathogens through close contact. This in turn leads to resistant pathogens that can have indirect and direct adverse public health effects on humans and animals [32]. Analysis of questionnaire responses revealed that veterinarians’ demographic data, including the year of graduation and age, did not influence the type of antimicrobials used. Most responses were from veterinarians who had recently graduated and were younger. This was in contrast to the findings from Australia, where the main sources of information (93%) that influenced antimicrobial prescribing decision-making for Australian dairy veterinarians were experienced clinicians [33]. In a recent study, where socio-demographic characteristics of veterinarians were considered, concerning antimicrobial prescription patterns, Servia-Dopazo et al., 2021 reported that the gender and age of the veterinarian do not affect the antimicrobial prescription pattern [34]. The majority of veterinarians surveyed indicated that they rely on literature such as treatment manuals for indications and administration when prescribing antimicrobials. This is consistent with reports from the United States, where peer-reviewed scientific literature and textbooks/drug manuals are the main sources of information for clinicians on antimicrobials [35]. Our study found that the use of Amoxy-Calv in small animal practice was the most commonly chosen by about 74.3% of respondents in Trinidad and 61.5% in Jamaica. These results are in line with previous reports [32,36], which identified amoxicillin-clavulanate as the most commonly prescribed antibiotics when infection was suspected but not confirmed. The results also confirmed the heavy use of other broad-spectrum antibiotics, such as penicillin, tetracyclines, sulfonamides, ceftriaxone, and aminoglycosides, which also contribute to the development of resistance in pathological microbial species such as *Salmonella* spp., *Staphylococcus* sp., and *Entero-Coccus* spp. that infect humans, livestock, pets, and livestock products [37,38,39,40]. It has been observed in this study that the culture and susceptibility (C&S) test is performed in both countries, but it is more utilized in Trinidad compared to Jamaica. Even though this susceptibility testing was performed, the majority of respondents used a broad-spectrum antimicrobial while waiting for test results. C&S testing significantly reduces the overuse of antibiotics, by isolating the specific causative agent of infection, allowing for the prescription of a narrow-spectrum antimicrobial. This reduces the recurrence of infections or the failure of antimicrobial use [41,42,43]. In Trinidad, the choice of the use of broad-spectrum antimicrobials might be primarily due to clients choosing a low-cost option that did not involve testing costs. Failure of clinics to have antibiotic policies in place may further lead to significant overuse of antimicrobials. A total of 96.2% of respondents in Trinidad and 82.9% in Jamaica reported having no such policy. A cross-sectional survey of the impact of antibiotic use guidelines in Denmark considered a subset of 63 guideline users, whose diagnostic habits were examined. Nineteen and 39% of respondents reported frequent culture and sensitivity (C&S) testing before treatment for pyoderma and urinary tract infection (UTI), respectively, and 68–84% reported C&S testing for poor response to treatment or recurrence of infection. In this regard, they attributed non-implementation of treatment recommendations to reliance on old prescribing practices and unavailability of the recommended drugs, and the main barriers to C&S testing were good experience with empirical treatment and the financial situation of owners [44]. This study also revealed that a client’s noncompliance to the treatment regimen effectively might lead to prolonged antimicrobial therapy, resulting in AMR. Veterinarians’ use of prophylactic antimicrobials during surgical procedures was another factor. A total of 36% of respondents always administered postoperative antimicrobials, and 22% of respondents frequently administered postoperative antimicrobials. The main reasons were the type of surgery (30% of respondents) and the duration of the surgery (28%). While prophylaxis is a valid reason for antimicrobial therapy, veterinarians should carefully consider whether it may be contributing to overuse, when other strategies may take the place of this routine, such as treating wounds of infected lesions before surgery or maintaining a sterile environment. A study of the prophylactic use of antibiotics by veterinarians during elective cesarean sections (CS) in Belgium showed a lack of consistency in antibiotic therapy during elective CS by rural veterinarians. While the choice of drug and duration of treatment were largely consistent with current guidelines, the same was not true for the dosage and route of administration. The majority of veterinarians injected antibiotics during or after surgery, while a minority administered antibiotics preoperatively, and most of them limited the duration of their antibiotic treatment to 1 day [45]. A cross-sectional study of the use of antimicrobials for surgical prophylaxis in cattle by veterinarians in Australia showed that a wide range of doses were used for surgical prophylaxis, with under dosing and inappropriate timing of administration being common reasons for inappropriate prophylactic treatment. The use of critical antimicrobials was very low [35].

Trinidad and Jamaica are considered “third world/developing countries”, since the drugs chosen by veterinarians are limited, unlike in developed countries. Therefore, the overuse of antimicrobials could be due to the supply and availability of antimicrobials in the health care system, and not necessarily as a preference. The outcome of this study might serve as a reference point to curtail the dangers and problems of AMR in veterinary and human medicine, by the authorities concerned imposing some measures and straightening the policies and guidelines on the use of antimicrobials.

In the Caribbean, there have been some initiatives to address antimicrobial resistance, ranging from public health to the agricultural sector. One such initiative was supported by the Pan American Health Organization, where microbiologists from across the Caribbean participated in a week-long intensive training on AMR. Another undertaking is the launch of a surveillance program in Jamaica, which grew out of a series of workshops during Antimicrobial Awareness Week 2020. The development of this program stemmed from the collaboration of the Jamaican Ministry of Agriculture and Fisheries with the Inter-American Institute for Cooperation in Agriculture (IICA). The main objective is to raise awareness of antibiotic resistance in food animals and to emphasize the importance of surveillance, using guidelines from the World Organization for Animal Health (WOAH), WHO, Advisory Group on Integrated Surveillance of Antimicrobial Resistance (AGISAR), and Codex. Previously, IICA issued a mission statement in 2016 in Caribbean countries including Trinidad and Jamaica, the sole purpose of which was to provide knowledge on the use of antibiotics in livestock production systems, to open pathways for future surveillance programs.

Based on the featured articles, as well as those referenced in the additional reading, it is clear that the Caribbean and its constituents are not mentioned in global research studies on the isolation and identification of resistant pathogens. In addition, while there are discussions of antibiotic resistance from both human and animal health perspectives, there is a lack of literature on the current surveillance and control of antibiotic use and enforcement programs. This lack of focus on the Caribbean underscores the need for initiatives against the overuse of antimicrobials and AMR to be explored in the future, especially as livestock production plays an important role in livelihoods in the Caribbean and as domestic animals are becoming increasingly popular. While the development of new pharmaceuticals is being investigated, it is important to maintain the currently available antimicrobials through controlled and necessary use in practice to reduce the incidence of AMR.

### 3.1. Limitations of the Study

The original approach was to compile a list of registered veterinarians in Trinidad and Jamaica and match them to associated clinics. To address this, a list of known small animal clinics throughout the country was compiled and these clinics were specifically contacted.

#### 3.1.1. Discrepancy in Clinic and Farm Lists

The list compiled for Trinidad included home visits and mobile veterinarians, while the list for Jamaica did not. Although the research was conducted separately for small animal clinics and farms, most small animal clinics and farms in both countries were mixed practices or had veterinarians working in mixed practices. This led to discrepancies in the listings.

#### 3.1.2. Information on Drug Use on Farms in Jamaica

Jamaican farms had non-disclosure agreements that prevented them from sharing information about their drug use. To address this problem, only small animal clinics and mixed practices in Trinidad and Jamaica were included in the data collection.

#### 3.1.3. Survey Dissemination

Originally, the survey was to be conducted by telephone, to accommodate the COVID-19 pandemic and associated protocols. This was eventually replaced by an online questionnaire distributed via email and social media, as this means proved to be more time-saving and convenient for both researchers and respondents. The difficulty, however, was that veterinarians responded late or not at all.

#### 3.1.4. Validity/Reliability/Risk of Bias of Questionnaires

Due to the subject matter of this study, there was an increased likelihood of bias in responding to the questionnaire. The disadvantages of the questionnaires were seen in the fact that respondents may have misunderstood the questions and were limited in providing additional information for the closed-response questions. There were no solutions to this.

## 4. Materials and Methods

### 4.1. Study Design and Questioner 

To facilitate COVID-19 limitations, a qualitative online questionnaire was conducted via a Google platform, based on purposive sampling intent and consisting of open-ended, multiple-choice, scaling, and dichotomous questions. Participants were asked a series of questions via a questionnaire, to collect relevant information about the habitual use of antimicrobials in their practice, whether it was a small animal or mixed practice. Several open-ended questions included the regions in which they practice and the names of the antimicrobials used for each species. Multiple-choice questions were used to explore other aspects, such as how antimicrobials are administered and the side effects of antimicrobials. There was a scaling question that focused on the duration of treatment for different organ systems. Dichotomous questions were used for information that could only be answered yes or no.

### 4.2. Sample Size

To determine the sample size, a list of small animal clinics in Jamaica and Trinidad were obtained through social media and their respective veterinary associations: the Jamaican Veterinary Medical Association (JVMA) and the Trinidad and Tobago Veterinary Association (TTVA). The sample size for each country was calculated using the formula below, with a 95% confidence level and a 5% margin of error. The results were 62 for Trinidad and 70 for Jamaica, with an average of 2 clinicians per clinic. The questionnaire was then distributed to all clinics and farms that could be reached through email and social media.

This medium allowed respondents to answer the survey at their leisure, and it ensured our safety by allowing limited exposure between our research group and the public. It also proved to be time and cost-efficient. It also allowed us to reach a larger population in a shorter time. The questionnaire itself was beneficial because it allowed for consistent data collection and gave respondents adequate time to answer the questions.

### 4.3. Statistical Analyses

Data were tabulated using Microsoft Excel for Microsoft 365 (Microsoft Inc., Redmond, WA, USA). Descriptive statistics were also implemented using Microsoft Excel. The Fisher exact test and chi-square test of independence were used to determine if there was any association between country and antibiotic usage. Statistical analyses were performed using the Vassar Stats package (http://vassarstats.net/ (accessed on 13 May 2022), and *p* values of <0.05 were considered statistically significant.

### 4.4. Ethical Consideration

For ethical reasons, respondents were assured that the data collected would be kept confidential.

## 5. Conclusions

Misuse of antimicrobials continues to be a major threat and leading to resistance. This study shows that broad-spectrum antimicrobials such as amoxicillin, penicillin/streptomycin combination, sulphonamides, and Fluoroquinolones are overused in small animal and mixed practices in both Trinidad and Jamaica and should be controlled, with appropriate monitoring of the adherence of clinicians to antimicrobial use guidelines. It has also highlighted the need for rigorous future monitoring and surveillance of antimicrobial use in the Caribbean, through proper recording of antimicrobial use, to control and contain antimicrobial resistance; as well as to ensure necessary use and reduce the global public health threat from a One Health perspective, taking into account human and animal health, and the threat of resistant pathogens also affecting general pharmaceutical care.

## Figures and Tables

**Figure 1 antibiotics-11-00885-f001:**
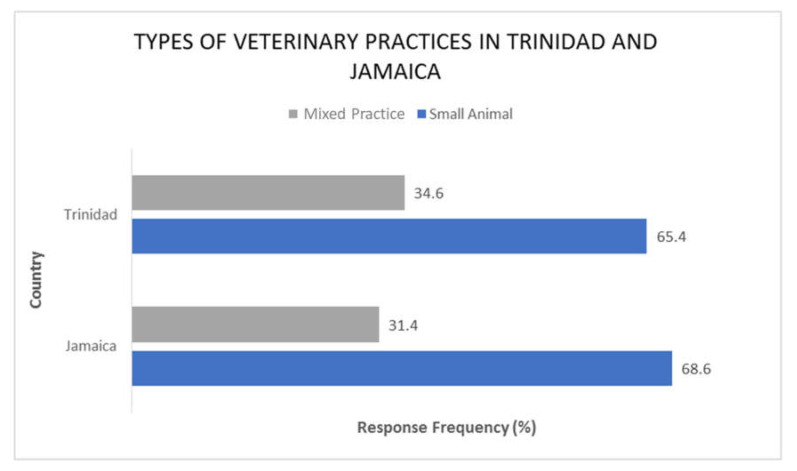
Showing the types of Veterinary practices in Trinidad and Jamaica.

**Figure 2 antibiotics-11-00885-f002:**
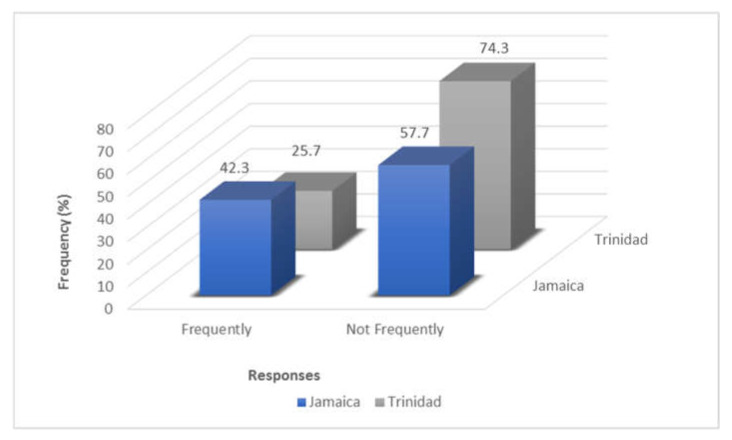
Comparison of how often combination therapy is utilized in Jamaica vs. Trinidad.

**Figure 3 antibiotics-11-00885-f003:**
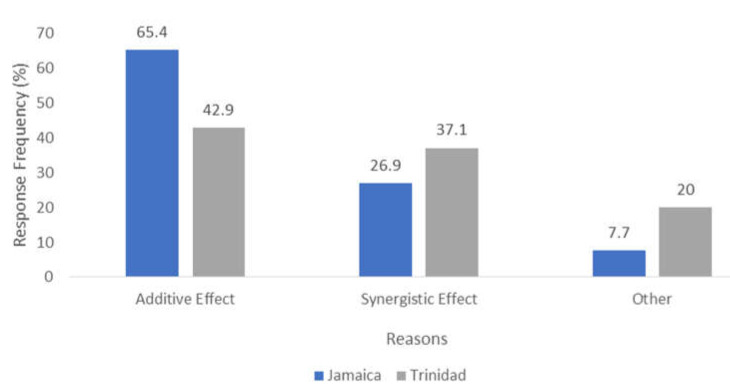
A bar chart showing the reasons for antimicrobial combination therapy in Trinidad vs. Jamaica.

**Figure 4 antibiotics-11-00885-f004:**
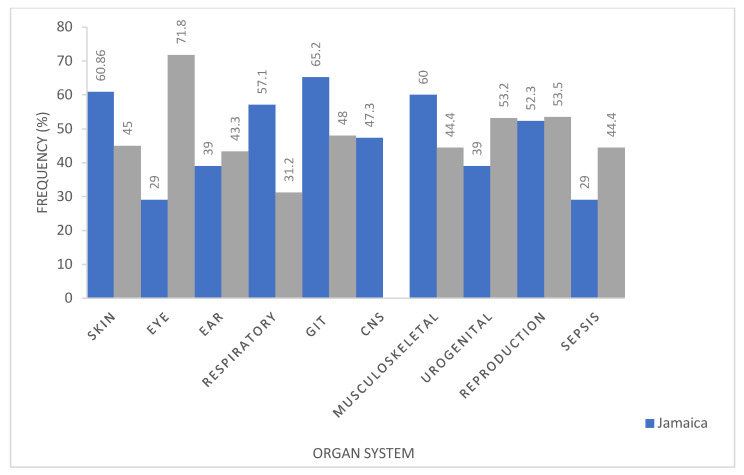
Comparison of the most common antimicrobials used for the different organ systems in veterinary practices in Jamaica vs. Trinidad.

**Figure 5 antibiotics-11-00885-f005:**
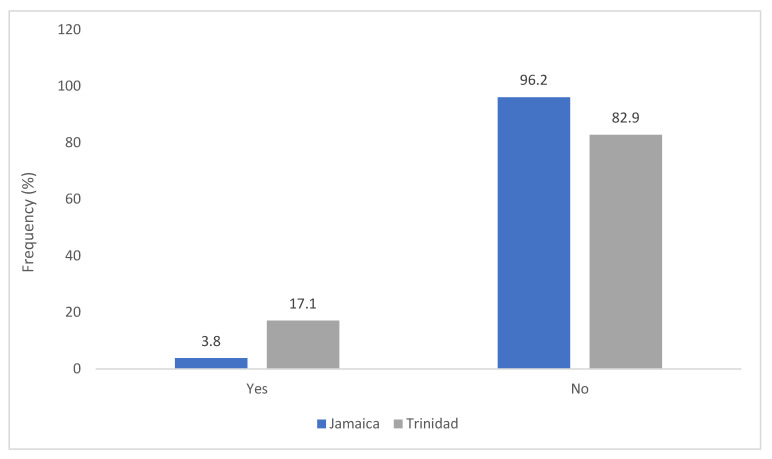
The presence of written antibiotic policies in clinics of Jamaica vs. Trinidad.

**Figure 6 antibiotics-11-00885-f006:**
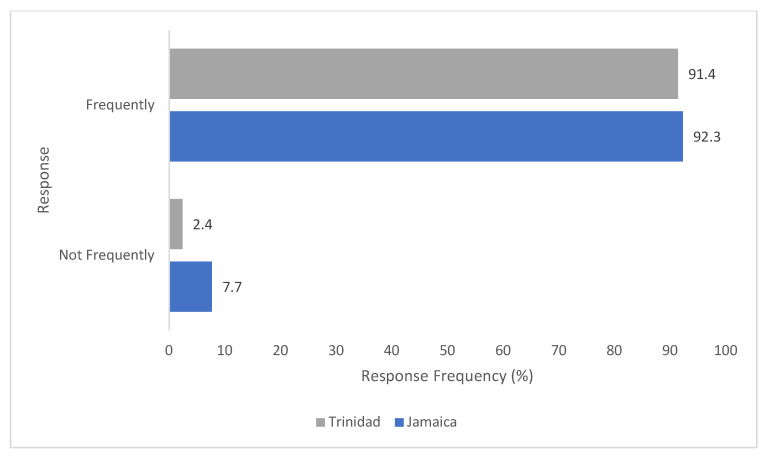
Comparison of how often antimicrobials are prescribed in Jamaica vs. Trinidad.

**Table 1 antibiotics-11-00885-t001:** Showing the Most commonly used Antimicrobial Agents in Small Animal Clinics and Mixed Animal Practices in Trinidad and Jamaica.

Type of Practice	Most Commonly Used Antimicrobial Agents	
**Small Animal Practice**	**Trinidad**	**Jamaica**
Dogs	Cephalexin (8; 25%), Trimethoprim-Sulfamethoxazole, (7; 21.9%), (Amoxicillin 11; 34.4%), Procaine Penicillin and Streptomycin (5; 15.6%), Doxycycline (21; 65.6%), Enrofloxacin, (12; 37.5%) Combikel (procaine benzylpenicillin, benzathine benzylpenicillin, dihydrostreptomycin sulfate (3; 9.4%), Ciprofloxacin, (5; 15.6%); Amoxicillin/ Clavulanic Acid, (23; 71.9%), Metronidazole (5; 15.6%); Terramycin (2; 6.3%), Gentamycin (2; 6.3%), Itraconazole (2; 6.3%) Ketoconazole (2; 6.3%)	Penicillin, (3;11.5%) Gentamicin, (7; 26.9%), Cephalosporins, (3; 11.5%) Amoxicillin Clavulanic (12; 46.2%) Acid, Trimethoprim Sulfa, (5; 19.2%) Cefadroxil, (2; 7.7% Cefuroxime (3; 11.5%) Metronidazole (6; 23.1%); Amoxicillin (14; 53.8%) Azithromycin (1; 3.8%), Ciprofloxacin (1; 3.8%), Doxycycline (5; 19.2%) Enrofloxacin, (4; 15.4%) Clindamycin (1; 3.8%), Ketoconazole (3; 11.5%), Avermectins (1; 3.8%) Fenbendazole (3; 11.5%);, Tetracyclines, (1; 3.8%), Ceftriaxone (1; 3.8%), Cefadroxil (2; 7.7%),
Cats	Amoxicillin (15; 46.9%), Amoxicillin/Clavulanic Acid, (25; 78.1%), Doxycycline (4; 12.5%), Metronidazole (3; 9.4%), Enrofloxacin (5; 15.6%), Trimethoprim-Sulfamethoxazole (2; 6.25%) Ciprofloxacin (2; 6.25%), Cephalexin (3; 9.4) Clotrimazole (1; 3.1%) Gentamycin (1; 3.1%),	Penicillins (2; 7.7%), Cefovecin (1; 3.8), Amoxicillin Clavulanic Acid, (10; 38.5%) Trimethoprim Sulfa, (5; 19.2%) Cefadroxil (1; 3.8%), Cefuroxime (3; 11.5%), Amoxicillin, (17; 65.9%) Ciprofloxacin(2; 7.7%) Doxycycline, (3; 11.5%) Enroflox-acin, (4; 15.4%), Gentamicin (4; 15.4%), Ceftriaxone, (1; 3.8%) Tetracyclines, (1; 3.8%) Ketoconazole, (1; 3.8%) Metronidazole (1; 3.8%)
**Mixed practices**		
Pigs	Procaine Penicillin and Streptomycin (1; 10%), Combikel (procaine benzylpenicillin, benzathine benzylpenicillin, dihydrostreptomycin sulfate) (2; 20%), Amoxicillin (1; 10%), Enrofloxacin (1; 10%)	Quinolones (1; 11.1%), Amoxicillin (2; 22.2%), Oxytetracycline (2; 22.2%) Neomycin-Tetracycline combination (1; 11.1%)
Poultry	Trimethoprim-Sulfamethoxazole (2; 20%) Piperazine (1; 10%), Amoxicillin (2; 20%)	Sulphonamide (2; 22.2%)
Sheep	Penstrep (Procaine Penicillin and Streptomycin) (5; 50%), Combikel (procaine benzylpenicillin, benzathine benzylpenicillin, dihydrostreptomycin sulfate) (2; 20%), Penbendazole (1; 10%); Sulphonamides (1; 10%)	Amoxicillin (2; 22.2%), Oxytetracycline,(1; 11.1%)
Cattle	Penstrep (Procaine Penicillin and Streptomycin), (6; 60%); Oxytetracycline (3; 30%); Combikel (procaine benzylpenicillin, benzathine benzylpenicillin, dihydrostreptomycin sulfate) (2; 20%)	Tetracycline, (1; 11.1%) Amoxicillin (3; 33.3%), Oxytetracycline, (1; 11.1%) Gentamicin, (1; 11.1%) Cephapirin (1; 11.1%)
Goats	Penstrep (Procaine Penicillin and Streptomycin), (6; 60%) Combikel (procaine benzylpenicillin, benzathine benzylpenicillin, dihydrostreptomycin sulfate), (2; 20%) Oxytetracycline,(2; 20%) Trimethoprim-Sulfamethoxazole, (1; 10%)	Amoxicillin, (4; 44.4% Oxytetracycline, (1; 11.1%) Trimethoprim-Sulfa (1; 11.1%)
Horses		Penicillins, (1; 11.1%) Gentamicin, (2; 22.2%) Oxytetracycline (1; 11.1%)
Porcupine	Enrofloxacin (1; 10%)	

**Table 2 antibiotics-11-00885-t002:** The association between country and the choice of antimicrobial agent in small animal clinics in Trinidad and Jamaica.

Drug	Country	Response		*p*-Value
		Yes	No	
Amoxicillin-Clavulanate	Trinidad Jamaica	26 16	9 10	0.43
Amoxicillin	Trinidad Jamaica	22 21	13 5	0.16
Sulphonamides	Trinidad Jamaica	13 8	22 18	0.81
Cephalosporin	Trinidad Jamaica	17 14	18 12	0.89
Fluoroquinolones	Trinidad Jamaica	13 14	22 12	0.30
Tetracycline	Trinidad Jamaica	21 10	14 16	0.16
Aminoglycoside	Trinidad Jamaica	7 11	28 15	0.11

**Table 3 antibiotics-11-00885-t003:** The association between country and choice of antimicrobial agent in mixed animal practices in Trinidad and Jamaica.

Drug	Country	No. of Response		*p*-Value
		Yes	No	
Penicillin combinations	Trinidad Jamaica	10 0	25 26	<0.01
Amoxicillin	Trinidad Jamaica	4 5	31 21	0.48
Sulphonamides	Trinidad Jamaica	3 3	32 23	1
Tetracyclines	Trinidad Jamaica	3 3	32 23	1
Aminoglycosides	Trinidad Jamaica	0 2	35 24	0.38
Fluoroquinolones	Trinidad Jamaica	13 9	22 17	0.84

## Data Availability

All the available data are included in the manuscript.

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
