# Peer review of "A Survey on the Rationale Usage of Antimicrobial Agents in Small Animal Clinics and Farms in Trinidad and Jamaica"

_antibiotics, 2022, doi:10.3390/antibiotics11070885_

Round 1
Reviewer 1 Report
The manuscript represented a very important issue, hence it's a time demand topic, I have a few suggestions for the author
1. In the abstract " Antibiotics are used indiscriminately in veterinary medicine" this line should be corrected, you should not claim a sector like veterinary without any proper documentation.
2. The introduction section should be more specific to explain the aim of the study. The author fails to describe the aim of the study properly.
3. In the table 1, table 2 and table 3: You should include the specific percentage of use of these drugs, only name is not enough to explain.
4. All figures are looking very colorful and color contrasts are not good. Try to use good contrast to make the figures and text within the figures more visible to the readers.
5. Try to discuss the present status of the antibiotic resistance in the study area and compare it to the other countries of the world and discuss how the present study will contribute to lessen the AMR.
Author Response
Response to Reviewer 1:
Thank you so much for your valuable comments and suggestions regarding our article entitled “A Survey on the Rationale Usage of Antimicrobial Agents in Small Animal Clinics and Farms in Trinidad and Jamaica”. Below are some point-to-point responses to your comments.
- In the abstract " Antibiotics are used indiscriminately in veterinary medicine" this line should be corrected, you should not claim a sector like veterinary without any proper documentation.
ANSWER: The word indiscriminately was removed and replaced by ‘extensively’
- The introduction section should be more specific to explain the aim of the study. The author fails to describe the aim of the study properly.
ANSWER: The aim of the study was captured both in the abstract(lines 19-20) and the introduction(lines 92-96)
- In the table 1, table 2, and table 3: You should include the specific percentage of use of these drugs, only name is not enough to explain.
ANSWER: Tables 1 and 3 were merged, number of the respondent and the percentage of responses for each drug has been included.
- All figures are looking very colorful and color contrasts are not good. Try to use good contrast to make the figures and text within the figures more visible to the readers.
ANSWER: The diagrams were replaced with much better ones in color and contrast
- Try to discuss the present status of the antibiotic resistance in the study area and compare it to the other countries of the world and discuss how the present study will contribute to lessen the AMR.
ANSWER: The was discussed (289-321)
Reviewer 2 Report
Table 1. It's unclear from the table shown what was the rate of usage of different antibiotics, or what is the criterion of the list made. If the statistics exist, it would be advisable to include them.
Line 82. Please include the number of specimens (n=).
Line 84. Please include the number of specimens (n=).
Line 102: "The most commonly used antimicrobial drugs across all species and practices were amoxicillin-clavulanic acid and amoxicillin (Table 3)." It is unclear from this statement, as well as from Table 3, whether amoxicilin-clavulanic acid and amoxicillin are the most commonly used antibiotic across species or are those most frequently used in every/all species. This should be clarified both in text and Table 3, similar to Table 1.
Line 106: "Most used microbiological analysis and antimicrobial susceptibility testing before antibiotic use...". How is 'most' defined? if there is a number, it should be clearly stated.
Line 109: "..respondents also reported that animals returned to their clinic because of antibiotic treatment failure with the primary reason being antimicrobial resistance..." The percentages and the total number of cases for each failure reason should be clearly stated in the text.
Line 121: "Most of the respondents (38.5%) also use postoperative antibiotics." What were the administration protocols? Prophylactic or therapeutic? Are data for the administration periods available? This should be cleared up and if available, the data shown in percentages of prophylactic vs therapeutic or in a similar manner.
Author Response
Response to Reviewer 2_Antibiotics
Thank you so much for your valuable comments and suggestions regarding our article entitled “A Survey on the Rationale Usage of Antimicrobial Agents in Small Animal Clinics and Farms in Trinidad and Jamaica”. Below are some point-to-point responses to your comments.
Comments and Suggestions for Authors
Table 1. It's unclear from the table shown what was the rate of usage of different antibiotics, or what is the criterion of the list made. If the statistics exist, it would be advisable to include them.
Answer: Tables 1 and 3 were merged, number of the respondent and the percentage of responses for each drug has been included.
Line 82. Please include the number of specimens (n=). Amended
Line 84. Please include the number of specimens (n=). Amended
Line 102: "The most commonly used antimicrobial drugs across all species and practices were amoxicillin-clavulanic acid and amoxicillin (Table 3)." It is unclear from this statement, as well as from Table 3, whether amoxicilin-clavulanic acid and amoxicillin are the most commonly used antibiotic across species or are those most frequently used in every/all species. This should be clarified both in text and Table 3, similar to Table 1.
Answer: This was clarified and amended both in the table and in the text
Line 106: "Most used microbiological analysis and antimicrobial susceptibility testing before antibiotic use...". How is 'most' defined? if there is a number, it should be clearly stated.
Answer: Amended
Line 109: "..respondents also reported that animals returned to their clinic because of antibiotic treatment failure with the primary reason being antimicrobial resistance..." The percentages and the total number of cases for each failure reason should be clearly stated in the text.
Answer: This was amended. See (line 124-127)
Line 121: "Most of the respondents (38.5%) also use postoperative antibiotics." What were the administration protocols? Prophylactic or therapeutic? Are data for the administration periods available? This should be cleared up and if available, the data shown in percentages of prophylactic vs therapeutic or in a similar manner.
Answer: All of administer them as post surgical management
Reviewer 3 Report
The manuscript entitled: "A Survey on the Rationale Usage of Antimicrobial Agents in Small Animal Clinics and Farms in Trinidad and Jamaica" submitted by Ismaila et al., discussed the pattern of antimicrobial use, appropriate pre-treatment, susceptibility testing, and the general distribution of antimicrobial use in both livestock and companion animals in two major CARICOM (Caribbean Community) countries, found very interesting in the concept and methodology of discussing a major worldwide health issue (Antimicrobial resistance-AMR).
The article in all is significant, being the first in the region and providing literature for future research in the same line; however, there are some flaws the authors should consider before submitting a revised version; this could be summarized in the following points:
1- In the abstract authors didn't mention the exact number of clinics & farms they've interviewed? I recommend indicating it.
2- In keywords, the authors wrote antibiotics & antimicrobial, considering one of them is better, in my opinion.
3- The introduction section is short; I recommend adding more statements regarding global AMR control strategies; for example, authors could benefit from these two references:
https://www.frontiersin.org/articles/10.3389/fpubh.2020.517964/full
https://www.ncbi.nlm.nih.gov/pmc/articles/PMC6854391/
4- In the introduction section, lines 36 - 37, the authors should consider putting just the abbreviation of WHO as they earlier mentioned the full name.
5- In the introduction section, lines 41 -43, I suggest that authors add to the text and specify how proximity & contact of companion animals (as they have the same meaning) with humans could help transfer antibiotic-resistance genes.
6- In the introduction section, lines 47 - 48, authors must consider giving Ref. for this statement & mention exactly what countries they mean by in line 48.
7- In the results section, lines 130 - 131, I suggest omitting the green label box of farms since it is not included in the study.
8- In the results section, lines 141 - 145, why the authors didn't mention the total P-value for both tables 4 & 5.
9- In the discussion section, lines 184 - 185, this statement needs re-write to make it more understandable; in this context, I suggest that the authors ask for a proofreading service from native English speakers aware of scientific language.
10- In the discussion section, sentences in lines 200 - 202 & lines 206 - 208 need Ref.
11- In the discussion section, line 212, UTI should be entirely written as it appears for the first time in the text.
12- In the discussion section, lines 219 - 220 needs language proofreading, my suggestion for authors like the same in point No. 9.
13- In the discussion section, line 253, the authors should consider changing OIE to its new acronym WOAH.
14- In line 352, Please consider mentioning the package version and date of your statistical program.
15- In Conclusion, If the authors summarize their findings in the light of the most used antibiotics and the Caribean Vets' perspective on AMR will be of much interest to readers.
Reviewer 4 Report
The manuscript by Ismaila is original, well written and interesting, aiming to identify the extent of the antimicrobial treatment in the Caribbean by assessing the use of antimicrobials in two major Caribbean islands: Trinidad and Jamaica. I support its possible publication after appropriate minor modifications as outlined below:
Line 13: please reject or rephrase this sentence, you dont have any proof for this statement
Line 75: please use predominantly lowercase in the title of the subchapters and tables
Line 76: Table 1 – I think that is not necessary to use the commercial name of the antimicrobials (e.g. Penstrep), if you defined the column “most commonly used antimicrobial agents”
Line 155: “no antibiotics” – please delete a supplementary space
Line 156: please insert the dote after the end of the sentence (Figure 3)
Line 303: “Questioner” – lowercase
Overall, I found in the reference list only 20 articles. As I know, the minimum requirement for full articles is “..should have more than 30 references.” (https://www.mdpi.com/journal/antibiotics/instructions). In this regard, the authors must find the way to improve the reference, list consulting and citing recently published articles, especially in Antibiotics journal (e.g. https://doi.org/10.3390/antibiotics10070846, https://doi.org/10.3390/antibiotics10121458) in order to increase the journal reputation and highlighting the over usage of drugs in veterinary medicine both food and non-food producing animals.
Author Response
Response to Reviewer 4_Antibiotics
Thank you so much for your valuable comments and suggestions regarding our article entitled “A Survey on the Rationale Usage of Antimicrobial Agents in Small Animal Clinics and Farms in Trinidad and Jamaica”. Below are some point-to-point responses to your comments.
Comments and Suggestions for Authors
The manuscript by Ismaila is original, well written and interesting, aiming to identify the extent of the antimicrobial treatment in the Caribbean by assessing the use of antimicrobials in two major Caribbean islands: Trinidad and Jamaica. I support its possible publication after appropriate minor modifications as outlined below:
Line 13: please reject or rephrase this sentence, you dont have any proof for this statement
ANSWER: The word indiscriminately was removed and replaced by ‘extensively’
Line 75: please use predominantly lowercase in the title of the subchapters and tables
ANSWER: Amended
Line 76: Table 1 – I think that is not necessary to use the commercial name of the antimicrobials (e.g. Penstrep), if you defined the column “most commonly used antimicrobial agents”
ANSWER: Amended
Line 155: “no antibiotics” – please delete a supplementary space
ANSWER: Amended
Line 156: please insert the dote after the end of the sentence (Figure 3)
ANSWER: Amended
Line 303: “Questioner” – lowercase
ANSWER: Amended
Overall, I found in the reference list only 20 articles. As I know, the minimum requirement for full articles is “..should have more than 30 references.” (https://www.mdpi.com/journal/antibiotics/instructions). In this regard, the authors must find the way to improve the reference, list consulting and citing recently published articles, especially in Antibiotics journal (e.g. https://doi.org/10.3390/antibiotics10070846, https://doi.org/10.3390/antibiotics10121458) in order to increase the journal reputation and highlighting the over usage of drugs in veterinary medicine both food and non-food producing animals.
ANSWER: Amended, reference sited
Round 2
Reviewer 3 Report
I was pleased to follow my suggested changes for the manuscript entitled “A Survey on the Rationale Usage of Antimicrobial Agents in Small Animal Clinics and Farms in Trinidad and Jamaica” by Ismaila et al., as now the quality of presentation looks more appropriate than the previous version. Still, minor language styles, Grammar checks, and missing references are required, especially for the discussion and conclusion parts, as I said in my earlier report (Points No. 9, 10, 11, 12, and the last one No. 15). However, I never saw a response for it, neither in the manuscript nor in the reply note file.
Rather than this, I find the article quite interesting for readers in regards to the significance of content, scientific soundness, and overall merit.
Author Response
Response to Reviewer3:
Thank you so much for your valuable comments and suggestions regarding our article entitled “A Survey on the Rationale Usage of Antimicrobial Agents in Small Animal Clinics and Farms in Trinidad and Jamaica”. Below are some point-to-point responses to your comments.
- In the abstract authors didn't mention the exact number of clinics & farms they've interviewed? I recommend indicating it.
Answer: Amended (see lines 20-25)
- In keywords, the authors wrote antibiotics & antimicrobial, considering one of them is better, in my opinion.
Answer: Amended (see lines 48)
3- The introduction section is short; I recommend adding more statements regarding global AMR control strategies; for example, authors could benefit from these two references:
https://www.frontiersin.org/articles/10.3389/fpubh.2020.517964/full
https://www.ncbi.nlm.nih.gov/pmc/articles/PMC6854391/
Answer: Amended (references included)
- In the introduction section, lines 36 - 37, the authors should consider putting just the abbreviation of WHO as they earlier mentioned the full name.
Answer: Corrected
- In the introduction section, lines 41 -43, I suggest that authors add to the text and specify how proximity & contact of companion animals (as they have the same meaning) with humans could help transfer antibiotic-resistance genes.
Answer: Amended
- In the introduction section, lines 47 - 48, authors must consider giving Ref. for this statement & mention exactly what countries they mean by in line 48.
Answer: Amended (references mentioned)
- In the results section, lines 130 - 131, I suggest omitting the green label box of farms since it is not included in the study.
Answer: Amended
8- In the results section, lines 141 - 145, why the authors didn't mention the total P-value for both tables 4 & 5.
Answer: Thank you for your suggestion. A paragraph has been added mentioning the p values. Lines 180-184……. (144-148 in the clean copy)
9- In the discussion section, lines 184 - 185, this statement needs re-write to make it more understandable; in this context, I suggest that the authors ask for a proofreading service from native English speakers aware of scientific language.
Answer: Thank you for your suggestion. The section was re-structured.. please see lines (188-191) highlighted yellow in the clean copy
10- In the discussion section, sentences in lines 200 - 202 & lines 206 - 208 need Ref.
Answer: Thank you for your suggestion. The section was re-structured and the references were cited please see lines (208-212) highlighted green in the clean copy
11- In the discussion section, line 212, UTI should be entirely written as it appears for the first time in the text.
Answer: Amended, please see line 218 (clean copy)
12- In the discussion section, lines 219 - 220 needs language proofreading, my suggestion for authors like the same in point No. 9.
Answer: Thank you for your suggestion. The section was re-structured.. please see lines (223-225) highlighted yellow in the clean copy
13- In the discussion section, line 253, the authors should consider changing OIE to its new acronym WOAH.
Answer: Amended (see line 261) clean copy
14- In line 352, Please consider mentioning the package version and date of your statistical program.
Answer: Thank you for your suggestion, the information has been added.
15- In Conclusion, If the authors summarize their findings in the light of the most used antibiotics and the Caribean Vets' perspective on AMR will be of much interest to readers.
Answer: Thank you for your suggestion. The findings were summarized in the light of the most used antibiotics and the Caribbean Vets' perspective on AMR see lines (347-357) highlighted green in the clean copy